# Effects of Temperature and Nutrition during the Larval Period on Life History Traits in an Invasive Malaria Vector *Anopheles stephensi*

**DOI:** 10.3390/insects14060543

**Published:** 2023-06-10

**Authors:** Nobuko Tuno, Thahsin Farjana, Yui Uchida, Mitsuhiro Iyori, Shigeto Yoshida

**Affiliations:** 1Laboratory of Ecology, Graduate School of Natural Science and Technology, Kanazawa University, Kanazawa 920-1192, Japan; 2Department of Parasitology, Faculty of Veterinary Science, Bangladesh Agricultural University, Mymensingh 2202, Bangladesh; thahsinfarjana@gmail.com; 3Laboratory of Vaccinology and Applied Immunology, School of Pharmacy, Kanazawa University, Kanazawa 920-1192, Japan

**Keywords:** *Plasmodium*, infection, life history traits, longevity, invasive species

## Abstract

**Simple Summary:**

*Anopheles stephensi* is an Asian malaria vector, and it has recently been discovered in Africa since 2012 and is still likely to expand its distribution there. *Anopheles stephensi* differs from other African malaria vectors in that it breeds in artificial containers in urban areas. Therefore, there is concern that this mosquito may spread malaria to areas where malaria has not been known. We raised larvae under variable conditions and provided blood infected with *Plasmodium berghei* at a constant temperature of 19 °C, the optimum temperature for growth of *P*. *berghei*. As a result, the survival rate of adults was affected by temperature and nutrient conditions at the larval stage. None of the mosquitoes that were raised at a high temperature (32 °C) and low nutrition conditions at the larval stage became infected with *P*. *berghei*. Adults’ infection rate and their larval experience were independent. At higher temperatures, only better-fed mosquitoes survived long enough to become infected. Mosquitoes’ wing length reflects their larval environment. Higher temperatures may reduce infective *A*. *stephensi*; however, larger-sized individuals can still be infected. We suggest that wing length can be a useful indicator to predict malaria risk in the field.

**Abstract:**

*Anopheles stephensi* is an Asian and Middle Eastern malaria vector, and it has recently spread to the African continent. It is needed to measure how the malaria parasite infection in *A*. *stephensi* is influenced by environmental factors to predict its expansion in a new environment. Effects of temperature and food conditions during larval periods on larval mortality, larval period, female wing size, egg production, egg size, adult longevity, and malaria infection rate were studied using a laboratory strain. Larval survival and female wing size were generally reduced when reared at higher temperatures and with a low food supply during the larval period. Egg production was not significantly affected by temperature during the larval period. Egg size was generally smaller in females reared at higher temperatures during the larval period. The infection rate of mosquitoes that fed on blood from malaria-infected mice was not affected by rearing temperature or food conditions during the larval period. Higher temperatures may reduce infection. *A*. *stephensi*; however, larger individuals can still be infective. We suggest that routinely recording the body size of adults in field surveys is effective in finding productive larval breeding sites and in predicting malaria risk.

## 1. Introduction

Malaria is one of the most serious human diseases, and even now it kills half a million people annually, mostly in sub-Saharan Africa. *Anopheles stephensi* Liston is a malaria vector mosquito that originated in Asia and the Middle East [1,2,3,4,5,6,7,8,9,10,11] and recently spread to the Horn of Africa and other African countries [12,13,14,15]. Different from other African malaria vector species, it occurs in urban areas where artificial containers are utilized as larval habitats [16]. Therefore, it is seriously concerned that it may cause the spread of malaria in urban areas or some other environments where malaria infection is not serious [13,17]. To prevent the expansion of malaria by this mosquito, it is crucial to evaluate the effects of environmental factors on its vectorial capacity.

For vector mosquitoes to become infective, they must survive at least the extrinsic inoculation period (EIP) of malaria parasites, a period required by parasites to develop from infection to sporozoites in the mosquito’s salivary glands [18,19]. EIP varies by parasite species and environmental conditions, but it is usually longer than 8 days [8]. In nature, the proportion of infective anopheline mosquitoes is only 1–2%; even more than 90% of people are infected [20], suggesting that only a few mosquitoes survive EIP. The longevity of female mosquitoes is affected by various factors such as the quantity and quality of blood meals, predation, and physical factors including temperature or humidity [5,8,21,22,23,24,25,26,27]. Body size is also related to longevity [28,29,30,31]. In mosquitoes as well as other holometabolic insects, the adult body size is determined by genetic factors and environmental conditions during larval periods [30]. In anopheline mosquitoes, several field studies have focused on the relationship between body size and survival [30,32], but there has been no consistency in the results; some reported positive correlations [28,31,33], while others did not [29]. These inconsistencies can be due to differences in study species, survey periods, or temperature regimes [34,35]. On the other hand, positive correlations are usually observed in laboratory studies [22,25,32,36]. If body size can be used as an indicator of longevity and vector competence, it is fortunate to be able to forecast the prevalence of malaria because body size is easily measured.

Temperature also affects the population dynamics of mosquitoes and their vector competence. It has been reported that the population fluctuation of *A*. *stephensi* is dependent on temperature rather than rainfall [1,14]. In addition, transmission of malaria by this mosquito occurred in the range of 17 and 35 °C, and the optimum temperature was 26–27 °C [2,3,23,24,26]. From these studies, it has been considered that global warming can suppress the prevalence of malaria caused by this mosquito [23]. However, there is also a prediction that the spreading of this mosquito will increase the risk of malaria infection [26].

This study aims to address how temperature and food conditions during the larval period affect their life history traits, larval period and survival, adult body size, longevity, fecundity, egg size, and their infection rate with *Plasmodium*. It is expected that low temperatures and a high food supply during the larval period will result in a larger female size and then a higher longevity and infection rate. This study will provide essential information to forecast the spread of this mosquito and its effect on the prevalence of malaria disease.

## 2. Materials and Methods

### 2.1. Animals and Malaria Parasite

The laboratory strain (SDA 500) of *Anopheles stephensi* was obtained from the Laboratory of Vaccinology and Applied Immunology of the School of Pharmacy of Kanazawa University. Before experiments, they were reared in our laboratory for a few generations at 27 °C under a photoperiod of 14 h light-10 h dark. Larvae were reared in plastic trays with a 2 cm depth of dechlorinated water using TetraMin (Spectrum Brands Japan Inc., Tokyo, Japan) as food. When pupae were formed, they were transferred to adult rearing cages (20 cm × 20 cm × 30 cm) with mesh walls. When adult mosquitoes emerged, a piece of cotton soaked with a 3% glucose solution was supplied in the cages as adult food. Three or four days after emergence, an anesthetized mouse was placed on a mesh ceiling to allow the mosquitoes to feed on blood. Blood-fed females were transferred into a plastic cup (2.6 cm in diameter and 5.5 cm in height) with wet cotton in the bottom and distilled water in the top, covered with a mesh sheet, and allowed to oviposit.

A transgenic strain of *Plasmodium berghei* (GFP luc) that expresses green fluorescent protein (EGFP) was obtained from Obihiro University of Agriculture and Veterinary Medicine. Whether mosquitoes are successfully infected or not is quickly determined by observing the expression of EGFP.

### 2.2. Effects of Larval Rearing Conditions on Larval Survival and Development and Female Wing Size

Blood-fed mosquitoes were introduced into a meshed cage (20 cm in width, 20 cm in height, and 30 cm in length) with cups filled with water and allowed to oviposit eggs for a day at 27 °C under a photoperiod of 14 h light-10 h dark. Eggs that oviposited were left in the cups under the same conditions and monitored for hatching every day. When larvae hatched, they were immediately transferred to 200 mL cups with a two cm depth of dechlorinated water and reared at 22, 27, 30, or 32 °C with a photoperiod of 14 h light-10 h dark under high and low food supplies. Under high food supply, 0.2 mg TetraMin was supplied to a cup per larva per day when larvae were young (the first and second instar stages), and 0.5 mg was supplied when they became older (the third and fourth instar stages). Under low food supplies, 0.05 and 0.1 mg of TetraMin were supplied per larva per day when larvae were young and old, respectively. At the time of the food supply, the water in the cups was renewed. These larvae were monitored for survival until pupation every day to determine mortality and larval periods. For each experimental regime, 10 cups with 20 larvae each were prepared. Sixteen hundred larvae were raised in total (4 temperatures × 2 diets × 20 larvae × 10 cups).

Pupae obtained from 10 cups were pooled, and then they were individually placed in plastic cases (2.6 cm in diameter and 5.5 cm in height) covered with a mesh sheet. A piece of wet cotton was placed on a mesh sheet as a water source for adult mosquitoes after emergence. The cases were kept at 27 °C for 14 h light-10 h dark. After adult emergence, they were successively maintained in these cases. When mosquitoes died, their wings were removed and measured for length (from the base to the tip) as a surrogate of body size. Measurement was done using the DP2-BSW software (Olympus, Tokyo, Japan) after digital images of wings were taken into a computer using a digital camera (Olympus DP25, Olympus, Tokyo, Japan) and a stereoscopic microscope (Olympus, Tokyo, Japan).

### 2.3. Effects of Larval Rearing Temperature on Egg Production and Egg Size

To examine egg production and egg size, a part of the pupae was randomly collected from 10 rearing cups, transferred to the above-mentioned adult rearing cages (40 mosquitoes per cage) with a piece of cotton soaked with 3% glucose solution, and kept at 27 °C for 14 h light-10 h dark. Four to thirteen days after adult emergence, an anesthetized mouse was placed on a mesh ceiling for 30 min to allow mosquitoes to feed on blood. Blood-fed mosquitoes were individually placed in the above-mentioned plastic cases (with a piece of cotton soaked with 3% glucose solution on the mesh) for egg maturation. A piece of wet cotton was placed at the bottom of the vial, which was covered by filter paper to serve as an ovipositioning substrate. Every female was checked daily for oviposition, and the date when it started was recorded. Mosquitoes were kept to be observed for another 3 days after they laid eggs for the first time. The number of eggs laid on the substrate was counted. In addition, 10 eggs were randomly selected from those oviposited by each female and measured for length in the same way as the measurement of wing length.

### 2.4. Effects of Larval Developmental Conditions on Plasmodium berghei Infection and Adult Survival

To prepare *Plasmodium berghei*-infected mice, a two-step infection protocol was adopted. In the first step, an 8-week-old mouse (the ICR strain obtained from Clea Japan Inc., Tokyo, Japan, ca. 30 g) was injected with 300 μL of phenylhydrazine to increase infection probability (PH process). After three days, the mouse was injected with blood infected by a transgenic strain (GFP luc) of *Plasmodium berghei*. Four days after blood infection, the proportion of red blood cells infected by *Plasmodium* was determined by observing mouse blood subjected to Giemsa staining. As a result, ca. 13% of red blood cells are infected. Then, whole blood was taken from the mouse after pentobarbital anesthesia. The mouse was killed by cervical dislocation while they were sleeping.

In the second step, collected blood (150 μL) was injected into four 9-week-old PH-processed ICR mice. Four days after the injection, the infection rate was checked by Gimsa staining; 10% of red blood cells were infected. The process was repeated when it was needed. In addition, exflagellation was observed to check the increase of *Plasmodium*. Five μL of blood was collected from the mouse tail, transferred into a cover glass, added to 15 μL of ookinate buffer (phenyl hydrazine hydrochloride, Sigma–Aldrich Co. Ltd., St. Louis, MO, USA), incubated at 19 °C for 10 min, and observed under a microscope. As a result, red-blood-cell condensations were observed, suggesting the occurrence of exflagellation. These four mice were used as blood meal sources for mosquitoes.

Mosquito larvae were reared at 27 or 32 °C under high and low food supplies in the same way as mentioned before. A temperature of 27 °C is optimum for egg-to-adult development, and 32 °C is the worst. After pupation, they were transferred to the adult rearing cages (200 pupae per cage) with a piece of cotton soaked with 3% glucose solution and kept at 19 °C [5] and 14 h light-10 h dark. Four or five days after emergence, adult mosquitoes were allowed to feed on blood from the *Plasmodium*-infected mouse anesthetized by intramuscular injection of 80 μL (30 μL/10 g of body weight of the mouse) of cocktails (ketalar 2 μL, xylazine 500 μL, saline 500 μL) for 50 min. Then, mosquitoes reared under the same conditions were divided into three groups. Mosquitoes of the first group were dissected 14 days post-feeding, and digital images of their midguts were taken into a computer using a digital camera (Olympus DP25, Olympus, Tokyo, Japan) and a fluorescent microscope (Axioskop2 ZEI22, Zeiss, Jena, Germany) to determine the presence or absence of oocysts in their midgut. If oocysts were present, their number on the surface of the midgut was counted. Mosquitoes of the second group were examined for the presence or absence of sporozoites in salivary glands 22 days post-feeding. However, it would be unrealistic for mosquitoes to go 22 days without feeding on blood. Therefore, mosquitoes of the third group were allowed to feed on blood again from the same mice 14 days after the first blood meal and examined for the presence and absence of sporozoites in salivary glands 6 days after the second blood meal. For mosquitoes in the second and third groups, survival was recorded 14 (for both groups) and 20 (for the third group) or 22 (for the second group) days after the first blood meal.

### 2.5. Statistics

Analyses were performed with a generalized linear model (GLM) or a generalized linear mixed model (GLMM) with a logit link function and a binomial error distribution when analyses were carried out on mortality, egg-hatching rate, and frequency of females with oocysts, but with an identity link function and a normal error distribution when analyses were carried out on developmental time, wing length, egg number, and egg length. Effects of temperature and food conditions on larval mortality and development time were analyzed by GLMM, including replicates as random effects, whereas those on female wing length were analyzed by GLM. In these analyses, the Akaike information criterion (AIC) was calculated for five models with the following fixed effects: (i) temperature (T), food conditions (F), and their interaction; (ii) T and F; (iii) only T; (iv) only F; and (v) none of them, and the best-fit model was selected.

Effects of temperature on egg production were analyzed by GLM, and those on egg length were analyzed by GLMM, including female identity as a random effect. AIC was calculated for the models with and without temperature as a fixed effect, and the better-fit model was selected. The frequency of female mosquitoes with oocysts in their midguts was analyzed by GLM, and model selection was carried out based on AIC. On the other hand, data on the frequency of mosquitoes with sporozoites in their salivary glands were not obtained for females at 32 °C under a low food supply. The frequency of mosquitoes with sporozoites was, therefore, compared by the χ^2^ test. The relationships between wing length and sporozoites presence in their salivary glands were analyzed by logistic regression.

The statistical analysis was performed using R v.3.6.1. (R Core Team 2019).

## 3. Results

### 3.1. Effects of Larval Rearing Conditions on Larval Mortality and Period and Female Wing Size

We investigated how mosquitoes’ life history traits are affected by food, temperature, and whether or not there is interaction. Table 1 shows larval mortality, larval development time, and female wing length at 22, 27, 30, and 32 °C under high and low food supplies (values were calculated with data pooled for 10 replicates). In the analysis of larval mortality, the model with temperature and food condition was selected according to the model selection based on AIC (Table 2), indicating that these two factors were independently affected. The development time was reduced with the increase in temperature in the range of 22–30 °C and increased a little at 32 °C. Therefore, the GLMM analysis was performed on the three datasets of “22–30 °C”, “22–32 °C” and “30–32 °C”. In the results, the model with temperature and food conditions was selected regardless of the datasets (Table 2). In the GLM analysis on wing length, the model including temperature, food condition, and their interaction was selected (Table 2).

### 3.2. Effects of Larval Rearing Temperature on Egg Production and Egg Size

We compared the number and size of eggs laid by females exposed to different rearing temperatures and food quantities during their larval stages. Table 3 shows the number and length of eggs oviposited by female mosquitoes that were reared at 22, 27, 30, and 32 °C under a high food supply during the larval period. In the GLM analysis of egg production, the null model was selected (Table 2). In the GLMM analysis of the effect of temperature on egg length, the model including temperature as a fixed effect was selected (Table 2).

### 3.3. Effects of Larval Rearing Conditions on Plasmodium berghei Infection and Adult Survival

We investigated how mosquitoes’ infection rates (oocyst stage and sporozoite stage) with *P*. *berghei* are affected by food and temperature experienced at the larval stages. Table 4 shows the percentages of mosquitoes with oocysts in their midguts, the number of oocysts counted in the positive midguts, and those with sporozoites in their salivary glands after females were fed with blood once or twice. In the analysis of the effects of temperature and food conditions on the frequency of mosquitoes with oocysts by GLMM, the null model was selected (Table 2). In addition, no significant difference was observed in the frequency of sporozoites in the salivary gland among the three conditions; high food supply at 27 °C, high food supply at 32 °C, and low food supply at 27 °C (χ^2^ test, *p* > 0.05). There were no mosquitoes that survived for 20 or 22 days under the conditions of a low food supply at 32 °C. There was no significant relationship between body size and sporozoite infection in adult salivary glands (logistic regression analysis, χ^2^ = 0.27, *p* > 0.6).

Figure 1A,B show the survival of female mosquitoes post-blood feeding. In mosquitoes that fed blood only once, survival at 14- and 22-days post-feeding was at least significantly lower in those reared at 32 °C under low food supply than in those reared under other conditions (Figure 1A, χ^2^ test with sequential Bonferroni correction, *p* < 0.05). In mosquitoes that fed blood twice, survival at 14 and 20 days after the first blood feeding was significantly lower in those reared at 32 °C than in those reared at 27 °C (Figure 1B, χ^2^ test with sequential Bonferroni correction, *p* < 0.05). When reared at 32 °C, survival was lower in mosquitoes reared under a low food supply than under a high food supply (χ^2^ test with sequential Bonferroni correction, *p* < 0.05), but the effect of food conditions was not significantly different in mosquitoes reared at 27 °C (χ^2^ test with sequential Bonferroni correction, *p* > 0.05).

## 4. Discussion

In this study, we aim to explore the effects of larval growing conditions on the life history traits of *A*. *stephensi*, a mosquito species that has recently spread to African countries [12,13,14,15]. The abundance of infective mosquitoes would be mainly dependent on (1) the abundance of mosquitoes that emerge from larval habitats, (2) the probability of blood-feeding from infected humans or animals, (3) the infection rate of mosquitoes that fed blood from infected humans or animals, and (4) the survival of mosquitoes for the extrinsic incubation period (EIC) of *Plasmodium* species [18]. In this study, larval survival decreased with increasing rearing temperatures, and none of the females raised at 32 °C under low food conditions survived the EIP of *P*. *berghei*. The wing length of these females was less than 3.0 mm, substantially shorter than the wings of females raised at lower temperatures or under high food conditions. GLMM revealed that the wing length was affected by larval rearing temperature, larval food conditions, and their interactions. The length of a mosquito’s wings can be a reliable and straightforward indicator to assess the risk of malaria in a given area. The EIP of *P*. *falciparum* varied from a minimum of 9.1 days to a maximum of 15.3 days, while the EIP of *P*. *vivax* varied from 8.0 to 24.3 days in *A*. *stephensi* [8]. Estimating the minimum wing length of *A*. *stephensi* to survive those EIPs within a possible temperature range will be helpful for the prevention of malaria risk.

In addition, fecundity decreased in mosquitoes that grew at higher temperatures, at least above 27 °C. Moreover, egg size was smaller in females that grew at higher temperatures. The reduction in egg size would result in lower larval survival. During a hot summer, *Aedes albopictus* (Skuse) produced thinner eggs without embryos and ceased laying eggs despite receiving blood meals [37]. Anopheles mosquitoes are known to require multiple blood meals for the first gonotrophic cycle, but the effects of high temperatures on egg quality remain unknown. To our knowledge, this is the first report of the small size of eggs laid by females who experienced high temperatures during their larval period, irrespective of larval food conditions. These characteristics (i.e., the reduction of larval survival, fecundity, and egg size) would act to retard the population increase. In contrast, the larval development period was reduced with temperature increases when the temperature was below 30 °C. Thus, the effects of temperature during the larval period on population increase are antagonistic. It was estimated that vectorial capacity reached its maximum at 26 or 27 °C for the combination of *A*. *stephensi* and *P*. *falciparum* [2,3]. A high temperature of 32 °C and low food supply yielded no infective mosquitoes, but mosquitoes survived long enough to become infective even at this temperature if the food was abundantly supplied in our experiment. Therefore, we emphasize that a combination of temperature information and the body size of adult females caught in the field can be good indicators for estimating malaria risk.

In conclusion, temperature and food conditions during the larval period substantially affect life history traits in *A*. *stephensi*. A high temperature and a low food supply during the larval period considerably shorten the longevity of adult female mosquitoes, yielding no infective mosquitoes. Whittaker et al. (2023) reviewed the literature on *A*. *stephensi* and concluded that it occurs in urban areas with high temperatures and that land use patterns, but not rainfall patterns, influence its abundance [14]. Malaria is not prevalent under high-temperature conditions, and the risk can be higher at lower temperatures [23,38,39]. Since we used mouse malaria and not human malaria, we cannot directly apply the present results to human malaria. However, if there is a suitable breeding place for larvae, the survival rate of adults will increase even at high temperatures. In the case of *An*. *gambiae sl*, we found suitable larval breeding water bodies near the sampling sites where large adults were common [40,41,42,43]. Routinely recording adults’ body size in field surveys effectively finds and determines the quality of larval breeding sites and predicts malaria risk.

## Figures and Tables

**Figure 1 insects-14-00543-f001:**
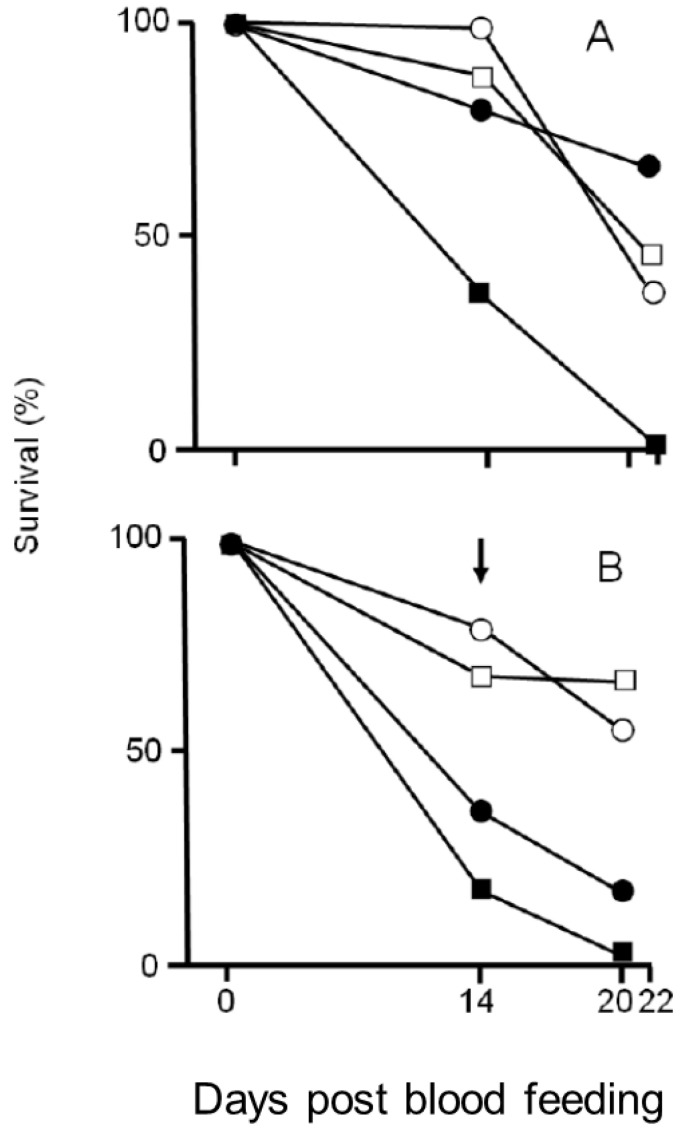
Survival female mosquitoes that were allowed to take blood meals from infected mice once (**A**) and twice (**B**), 14 and 20 (or 22) days after the first blood meal. Open circle (reared at 27 °C under high food supply during the larval period); open square (reared at 27 °C under low food supply); closed circle (at 32 °C under high food supply); closed square (at 32 °C under low food supply). The arrow indicates the second blood meal.

**Table 1 insects-14-00543-t001:** Larval mortality, larval period, and female wing length of *A*. *stephensi* when reared at 22, 27, 30, and 32 °C under high and low food supply during the larval period.

Food Supply	Temp. (°C)	Larval Mortality		Larval Period (Days)		Wing Length (mm)	
		N	%	N	Mean ± SD	N	Mean ± SD
High	22	200	6.5	187	12.37 ± 1.20	30	3.44 ± 0.31
	27	200	5.5	189	9.53 ± 1.18	28	3.54 ± 0.18
	30	200	9.5	182	8.80 ± 1.27	13	3.29 ± 0.12
	32	200	14.5	172	9.43 ± 1.26	11	3.19 ± 0.12
Low	22	200	11.0	179	18.71 ± 1.83	21	3.35 ± 0.15
	27	200	10.0	182	15.48 ± 2.87	24	3.11 ± 0.25
	30	200	12.5	175	14.38 ± 3.26	10	2.96 ± 0.12
	32	200	28.5	145	15.43 ± 2.08	3	2.74 ± 0.02

**Table 2 insects-14-00543-t002:** Akaike Information Criterion values in the GLM or GLMM analyses on life history traits, (1) model with temperature (T), food condition (F) and their interaction as fixed factors, (2) model with T and F, (3) model with T, (4) model with F, and (5) model with none of them. The minimum value is suggested in bold.

Traits	Model				
	**1**	**2**	**3**	**4**	**5**
Larval mortality	1136	**1134**	1140	1142	1147
Larval development time					
22–30 °C	4281.8	**4278.2**	4370.3	4322.7	4380.3
22–32 °C	5516.7	**5515**	5638.1	5559	5648.8
30–32 °C	2722	**2720.6**	2784.6	2722.8	2783.5
Female wing length	**−13.8**	−7	32.4	20.6	50.5
Egg number	-	-	258.3	-	**256.8**
Egg length	-	-	**2261.4**	-	2263.4
Frequency of females with oocysts	20.9	22.5	20.8	20.7	**18.9**

**Table 3 insects-14-00543-t003:** The number of eggs oviposited by female mosquitoes reared at 22, 27, 30, and 32 °C under high food supply during the larval period and the length of eggs produced.

Temp. (°C)	No. of Eggs Oviposited		Egg Length (µm)	
	N	Mean ± SD	N	Mean ± SD
22	5	65.5 ± 5.2	47	559.7 ± 35.9
27	8	136.8 ± 64.2	80	558.6 ± 29.2
30	5	113.0 ± 17.0	49	525.9 ± 26.8
32	6	93.5 ± 27.7	60	528.1 ± 51.4

**Table 4 insects-14-00543-t004:** Percentage of mosquitoes with oocysts and number of oocysts in positive individuals in their midguts and those with sporozoites in their salivary glands. Mosquitoes were reared at 27 and 32 °C under high and low food supplies and allowed to feed on mice’s blood once or twice.

		One Blood Meal									Two Blood Meals	
Food Supply	Temp. (°C)	Oocysts			N Oocyst			Sporozoits			Sporozoits	
		N	(%)		N Positive	Mean ± SE		N	(%)		N	(%)
High	27	12	25		3	13.8 ± 7.7		30	33		39	46
	32	17	71		12	14.7 ± 9.5		28	39		17	29

Low	27	16	50		8	2.8 ± 0.6		28	57		40	23
	32	16	31		5	13.4 ± 3.4		-	-		-	

## Data Availability

All data are shown in the manuscript.

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
