# Peer review of "Effects of Temperature and Nutrition during the Larval Period on Life History Traits in an Invasive Malaria Vector Anopheles stephensi"

_insects, 2023, doi:10.3390/insects14060543_

Round 1
Reviewer 1 Report
In this manuscript, Dr. Tuno and colleagues report the effects of temperature and nutrition during larval stages in life history traits and infection of the mosquito Anopheles stephensi. The overall goal is to have a better understanding of how environmental changes during larval development can affect population structure and the ability to transmit the malaria pathogen.
The paper is relevant however it needs improvement before publication. The English needs extensive editing throughout the paper, as some parts is not possible to understand what the authors want to communicate. Introduction is missing key literature including papers that have performed similar experiments with the same mosquito species, and the discussion is superficial and is also missing key work for comparisons. Please find below my comments/suggestions, in the hope of enhancing the quality of this paper.
Italicize Anopheles stephensi throughout the manuscript and although not incorrect instead of writing that mosquitoes sucked blood, I suggest changing to blood fed, also throughout that paper.
Simple summary:
Line 18: place a coma after “temperature” and remove “and” after it; place another coma after “conditions”: rearing temperature, nutritional conditions, and adult survivorship to be ….
Line 19 – 23: The results stated in those lines need to be specific. The way it is written creates doubt and uncertainty regarding the findings on the paper.
Abstract:
Line 24: Anopheles stephensi is still an Asian malaria vector, however it has spread to African countries.
Line 25: …to measure how the malaria parasite infection in ….
Introduction:
Line 43: cite the most recent literature from WHO.
Line 48: superficial definition of vectorial capacity. Needs improvement.
Line 50-51: Not sure what the authors meant on that phrase. Needs to be rephrased.
Line 54: …time of mosquitoes, further we would be able to reduce….
Line 55: …female mosquitoes is affected by various factors….
Line 57: …is also related to longevity….
Line 69: …caused by this species.
Methods:
Line 85: …to quickly enable observation and count….
Simple questions: The blood-fed mosquitoes were placed in cups containing water in the bottom of the or containers containing water (for oviposition) were placed inside the cups? (line87-88)
The larvae were transferred to 200ml cups or containers?
Is there any particular reason why the low food supply group had 4X less food in the early stages and 5X less on the older stages in comparison to the high food supply group?
Line 126: …. with blood infected by GFP luc…. (The GFP luc was explained in the beginning of materials and methods)
Line 134-135: This part needs to be redone.
Line 136: please describe ookinete buffer.
Line 154: Why the number of oocysts were counted if the data is not presented in the manuscript? Only infection rate is presented on table 4.
Line 156 and 158: What do the authors mean by whole body? It needs a better description.
Results:
On all subtitles a brief introduction to the results is appreciated. The way that each one starts seems like a table legend. Overall, the results are hard to read and are robotically written and a good revision is needed.
Table 1: Fix typos in the column titles: Food supply and enlarge larval mortality.
Line 234: fix typos: Oviposited and mosquito
Discussion:
Line 240-242; 246; 253-256; 275-277: Rephrase. Hard to understand.
Line 262: Table 4
The discussion is superficial and the results are not properly discussed. A good revision is needed.
Acknowledgements: Italicize Plasmodium
The entire manuscript needs to have the English revised.
Author Response
In response to Reviewer 1,
We appreciate your time in reading the manuscript and for giving thoughtful comments.
Here, we write our answers to the reviewer's comments in blue.
Comments and Suggestions for Authors
In this manuscript, Dr. Tuno and colleagues report the effects of temperature and nutrition during larval stages in life history traits and infection of the mosquito Anopheles stephensi. The overall goal is to have a better understanding of how environmental changes during larval development can affect population structure and the ability to transmit the malaria pathogen.
The paper is relevant however it needs improvement before publication. The English needs extensive editing throughout the paper, as some parts is not possible to understand what the authors want to communicate. Introduction is missing key literature including papers that have performed similar experiments with the same mosquito species, and the discussion is superficial and is also missing key work for comparisons. Please find below my comments/suggestions, in the hope of enhancing the quality of this paper.
Reply. Thank you for your suggestions.
We added more references in the Introduction.
Italicize Anopheles stephensi throughout the manuscript and although not incorrect instead of writing that mosquitoes sucked blood, I suggest changing to blood fed, also throughout that paper.
Reply. Thank you. We corrected those mistakes.
Simple summary:
Line 18: place a coma after “temperature” and remove “and” after it; place another coma after “conditions”: rearing temperature, nutritional conditions, and adult survivorship to be ….
Reply. We corrected.
Line 19 – 23: The results stated in those lines need to be specific. The way it is written creates doubt and uncertainty regarding the findings on the paper.
Reply. Thank you for your suggestions. We tried to describe more specifically.
Abstract:
Line 24: Anopheles stephensi is still an Asian malaria vector, however it has spread to African countries.
Reply. We corrected.
Line 25: …to measure how the malaria parasite infection in ….
Reply. We corrected.
Introduction:
Line 43: cite the most recent literature from WHO.
Reply. Thank you. We corrected.
Line 48: superficial definition of vectorial capacity. Needs improvement.
Reply. Thank you. We rewrote the definition.
Line 50-51: Not sure what the authors meant on that phrase. Needs to be rephrased.
Reply. Thank you. We rewrote the part.
Line 54: …time of mosquitoes, further we would be able to reduce….
Reply. Thank you. We rewrote the part.
Line 55: …female mosquitoes is affected by various factors….
Reply. Thank you. We rewrote the part.
Line 57: …is also related to longevity….
Reply. Thank you. We rewrote the part.
Line 69: …caused by this species.
Reply. Thank you. We rewrote the part.
Methods:
Line 85: …to quickly enable observation and count….
Reply. Thank you. We rewrote the part.
Simple questions: The blood-fed mosquitoes were placed in cups containing water in the bottom of the or containers containing water (for oviposition) were placed inside the cups? (line87-88)
Reply. Thank you. We rewrote the part. Blood-fed mosquitoes were introduced into a meshed cage (20cm in width, 20cm in height, 30cm in length) with ovicups with water and allowed to oviposit eggs for a day at 27 °C under a photoperiod of 14 h light-10 h dark.
The larvae were transferred to 200ml cups or containers?
Reply. The larvae were transferred to 200ml cups.
Is there any particular reason why the low food supply group had 4X less food in the early stages and 5X less on the older stages in comparison to the high food supply group?
Reply. There is no reason. We measured the food weight that we fed the larvae to make body size differences bigger but with less mortality.
Line 126: …. with blood infected by GFP luc…. (The GFP luc was explained in the beginning of materials and methods)
Reply. Thank you, we deleted the explanation.
Line 134-135: This part needs to be redone.
Reply. Thank you. We repeated the check to find infection rate fell within 5 to 10%. We add the sentence.
Line 136: please describe ookinete buffer.
Reply. We add the information.
Line 154: Why the number of oocysts were counted if the data is not presented in the manuscript? Only infection rate is presented on table 4.
Reply. Thank you. We add the information in Table 4.
Line 156 and 158: What do the authors mean by whole body? It needs a better description.
Reply. Thank you for your question. Whole body means that sporozoites found in the body out of salivary glands. As it may be confusing, we deleted the data of whole body as we did not know the significance in terms of infection.
Results:
On all subtitles a brief introduction to the results is appreciated. The way that each one starts seems like a table legend. Overall, the results are hard to read and are robotically written and a good revision is needed.
Reply. Thank you for your suggestion. We add a sentence to explain the paragraph in the head.
Table 1: Fix typos in the column titles: Food supply and enlarge larval mortality.
Reply. Thank you for your comments. We corrected.
Line 234: fix typos: Oviposited and mosquito
Reply. Thank you for your comments. We corrected.
Discussion:
Line 240-242; 246; 253-256; 275-277: Rephrase. Hard to understand.
Line 262: Table 4
Reply. Thank you for your comments. We rewrote the discussion part and asked for a professional English editing service.
The discussion is superficial and the results are not properly discussed. A good revision is needed.
Reply. Thank you for your comments. We rewrote the discussion part and asked for a professional English editing service.
Acknowledgements: Italicize Plasmodium
Reply. Thank you for your comments. We corrected.
Comments on the Quality of English Language
The entire manuscript needs to have the English revised.
Reply. Thank you for your comments. We send our manuscript to a professional editor for proof reading.
We greatly appreciate the reviewers’ time, significant concerns, and constructive comments on the study.
Reviewer 2 Report
There is a vast literature on thermal impacts on the ecology of An. stephensi that is not referenced. There is also a great deal of information on the current spread and impact of An. stehensi that should be referenced, https://www.mesamalaria.org/mesa-track/deep-dives/anopheles-stephensi and https://apps.who.int/malaria/maps/threats/. I also object to the use of the collective "Vectorial Capacity" when the paper is only addressing longeviety assessed in the lab.

English is very good. Only issue is use of the word "blood sucking" rather than "blood feeding"
Author Response
In response to Reviewer 2.
We appreciate your time in reading the manuscript and for giving great comments.
Here, we write our answers to the reviewer's comments.
There is a vast literature on thermal impacts on the ecology of An. stephensi that is not referenced. There is also a great deal of information on the current spread and impact of An. stehensi that should be referenced, https://www.mesamalaria.org/mesa-track/deep-dives/anopheles-stephensi and https://apps.who.int/malaria/maps/threats/. I also object to the use of the collective "Vectorial Capacity" when the paper is only addressing longeviety assessed in the lab.
Reply. Thank you for your suggestions.
We followed and added more references in the Introduction.
We change the word "Vectorial Capacity" to longevity or life history traits.
Commented [A1]: I would change the title to read
"…..period on laboratory reared longevity of an invasive …"
as you are only looking at one parameter of the VC model.
Reply. Thank you for your suggestions.
We changed the title not to use VC.
Commented [A2]: No. It was first detected in Djibouti in 2012: Faulde MK, Rueda LM, Khaireh BA. First record of the Asian malaria vector Anopheles stephensi and its possible role in the resurgence of malaria in Djibouti, Horn of Africa. Acta Trop. 2014 Nov;139:39-43. doi: 10.1016/j.actatropica.2014.06.016. Epub 2014 Jul 5. PMID: 25004439.
Reply. Thank you for your comment.
We corrected.
Commented [A3]: Not "East Africa". Say "Horn of Africa" it appears to have spread from Djibouti to Ethiopia Carter TE, Yared S, et al. Genetic diversity of Anopheles stephensi in Ethiopia provides insight into patterns of spread. Parasit Vectors. 2021 Dec 11;14(1):602. doi: 10.1186/s13071-021- 05097-3. PMID: 34895319; PMCID: PMC8665610. and Sudan Abubakr M, et al. The Phylodynamic and Spread of the Invasive Asian Malaria Vectors, Anopheles stephensi, in Sudan. Biology (Basel). 2022 Mar 7;11(3):409. doi: 10.3390/biology11030409. PMID: 35336783; PMCID: PMC8945054. Authors should also note the WHO Malaria Threats Map that collates data on current extent of spread https://apps.who.int/malaria/maps/threats/
Reply. Thank you for your comment.
We updated the information.
Commented [A4]: No. The only documented malaria outbreaks due to An. stephensi are Djibouti and likely Dira Dawa in Ethiopia https://assets.researchsquare.com/files/rs2709856/v1/6a9945e2-2b78-4718-8ded90e26b26964b.pdf?c=1679414000
Reply. Thank you for your comment.
We deleted the word of outbreak.
Commented [A5]: I am surprised there are no references to the extensive An. stephensi work in Chennai through the ICEMR grants, e.g. Cator, L.J., Thomas, S., Paaijmans, K.P. et al. Characterizing microclimate in urban malaria transmission settings: a case study from Chennai, India. Malar J 12, 84 (2013). https://doi.org/10.1186/1475-2875-12-84
Reply. Thank you for your comment.
We added the reference of Cator et al. 2013.
Commented [A6]: Not exactly. You should reference Garrett-Jones (1964) who came up with the term: GarrettJones, C. "Prognosis for interruption of malaria transmission through assessment of the mosquito's vectorial capacity." Nature 204 (1964): 1173-1175.
Reply. Thank you for your comment.
We changed the reference to Garrett-Jones 1964.
Commented [A7]: Especially in the case of the very zoophilic An. stephensi you should also address the human biting rates, in the Vectorial Capacity model, "ma"
Reply. Thank you for your comment. If my understanding is correct, in the field survey the parameter can be a variable. We see the parameter as a fixed value in Anopheles stephensi in this study.
Commented [A8]: I wish you would stop saying VC - you are looking at just one parameter, longevity, not the overall vectorial capacity.,
Reply. Thank you for your comment. We did not use vectorial capacity.
Commented [A9]: Say bloodmeal, not sucking
Reply. Thank you for your comment.
We changed the expression.
Commented [A10]: Now in Nigeria and Ghana
Reply. Thank you for your comment.
We changed the word East Africa to African countries.
Commented [A11]: Here say feeding not sucking
Reply. Thank you for your comment.
We changed sucking to feeding.
Commented [A12]: It is not just high temperature, but more important lower humidity that shortens adult life
Reply. Thank you for your comment.
It must be true though we did not mention lower humidity which was out of our control in the experiment.
Commented [A13]: Where is there any evidence that larval rearing impacts blood feeding behavior, e.g. the HBI?
Reply. Thank you for your comment.
In Grech et al 2007, they reported that “However, parental effects were influential in determining the fecundity of daughters. Counter-intuitively, daughters of parents reared in low food conditions produced larger egg clutches than daughters of parents reared in high food conditions. Offspring reared in low food conditions took larger blood meals if their parents had also experienced a low food environment.”
Commented [A14]: No: there is an upper thermal limit to sporogony - see the work of Matt Thomas on An. stephensi in India
Reply. Thank you for your comment. In the manuscript, the word temperature represents the larval rearing temperature of the feeding mosquitoes but not the air temperature when mosquitoes took blood meals from the infected mouse. We rewrote the part to avoid confusion.
Commented [A15]: No: the predicted EIP of P. falciparum varies from a minimum of 9.1 days to a maximum of 15.3 days, while the EIP of P. vivax varies from 8.0 to 24.3 days depending on the microclimate See: Thomas S, Ravishankaran S, Justin NAJA, Asokan A, Kalsingh TMJ, Mathai MT, Valecha N, Montgomery J, Thomas MB, Eapen A. Microclimate variables of the ambient environment deliver the actual estimates of the extrinsic incubation period of Plasmodium vivax and Plasmodium falciparum: a study from a malaria-endemic urban setting, Chennai in India. Malar J. 2018 May 16;17(1):201. doi: 10.1186/s12936-018-2342-1. PMID: 29769075; PMCID: PMC5956829.
Reply. Thank you for your comment. We rewrote the relevant part to avoid confusion as EIP is variable.
Commented [A16]: Don't think you can make sweeping conclusions from this study
Reply. Thank you for your comment. But we concluded that adult performance reflects their larval environment. We do not think this is a sweeping conclusion.
Commented [A17]: Ref: Fazeli-Dinan, M., Azarnoosh, M., Ö zgökçe, M. et al. Global water quality changes posing threat of increasing infectious diseases, a case study on malaria vector Anopheles stephensi coping with the water pollutants using age-stage, two-sex life table method. Malar J 21, 178 (2022). https://doi.org/10.1186/s12936-022-04201- x
Reply. Thank you for your comment. We add the great study in the references.
We thank the reviewers invaluable time and productive comments to improve our manuscript.
Round 2
Reviewer 2 Report

I may have done too much editing of the english, but you need a good copy-editor for the final. But much improved, thanks.
Author Response
Dear Reviewer,
Thank you for your patient correction of the manuscript.
We followed the reviewer's correction.
There were many mistakes in the second version, we rephrased many parts shown in red.
We appreciate your time and constructive correction for the manuscript.
All authors.